# Establishment and stability of the latent HIV-1 DNA reservoir

**Johanna Brodin[1], Fabio Zanini[2†], Lina Thebo[1], Christa Lanz[2], Göran Bratt[3,4], Richard A Neher[2], Jan Albert[1,5*]**

[1]Department of Microbiology, Tumor and Cell Biology, Karolinska Institute, Stockholm, Sweden; [2]Max Planck Institute for Developmental Biology, Tübingen, Germany; [3]Department of Clinical Science and Education, Stockholm South General Hospital, Stockholm, Sweden; [4]Venhälsan, Stockholm South General Hospital, Stockholm, Sweden; [5]Department of Clinical Microbiology, Karolinska University Hospital, Stockholm, Sweden

**Abstract** HIV-1 infection cannot be cured because the virus persists as integrated proviral DNA in long-lived cells despite years of suppressive antiretroviral therapy (ART). In a previous paper (*Zanini et al, 2015*) we documented HIV-1 evolution in 10 untreated patients. Here we characterize establishment, turnover, and evolution of viral DNA reservoirs in the same patients after 3–18 years of suppressive ART. A median of 14% (range 0–42%) of the DNA sequences were defective due to G-to-A hypermutation. Remaining DNA sequences showed no evidence of evolution over years of suppressive ART. Most sequences from the DNA reservoirs were very similar to viruses actively replicating in plasma (RNA sequences) shortly before start of ART. The results do not support persistent HIV-1 replication as a mechanism to maintain the HIV-1 reservoir during suppressive therapy. Rather, the data indicate that DNA variants are turning over as long as patients are untreated and that suppressive ART halts this turnover.

*For correspondence: Jan. Albert@ki.se

**Present address:** [†]Stanford University, Stanford, CA, United States

## Introduction

Combination antiretroviral therapy (ART) has had significant effects on the morbidity and mortality associated with human immunodeficiency virus type 1 (HIV-1) infection. ART very effectively suppresses active virus replication, but it cannot eradicate the infection because HIV-1 persists as integrated proviral DNA in long-lived cells that constitute a virus reservoir. Latently infected resting memory CD4+ T-lymphocytes (memory CD4 cells) represent the most solidly documented HIV-1 reservoir (*Eriksson et al., 2013*; *Chun et al., 1997*, *1995*). Fully functional integrated HIV-1 proviruses are present in a small fraction of memory CD4 cells. These cells do not produce virus when they are in a resting state, but can be induced to produce virus upon activation in vitro and in vivo (*Eriksson et al., 2013*; *Chun et al., 1995*, *1997*; *Massanella and Richman, 2016*). The activation state of the infected cell and the viral encoded Tat feedback loop jointly determine latency and virus production (*Razooky et al., 2015*; *Rouzine et al., 2015*).

Because of their importance to development of a cure for HIV-1 infection, many methods to quantify HIV-1 reservoirs have been developed. The quantitative virus outgrowth assay (QVOA) has been the 'gold standard' (*von Stockenstrom et al., 2015*; *Massanella and Richman, 2016*; *Bruner et al., 2015*), but recent studies have revealed that this assay greatly underestimates the true size of the functional reservoir (*Ho et al., 2013*; *Bruner et al., 2016*). In contrast, PCR-based assays overestimate the size of the functional reservoir because they cannot distinguish between replication-competent and defective viral genomes (*von Stockenstrom et al., 2015*; *Massanella and Richman, 2016*; *Bruner et al., 2015*, *2016*). Many defective proviruses contain large internal

deletions (*Ho et al., 2013*; *Sanchez et al., 1997*). Defective proviruses also result from APOBEC editing, which induces G-to-A hypermutation (*Yu et al., 2004*; *Kieffer et al., 2005*; *Stopak et al., 2003*; *Bruner et al., 2016*).

The HIV-1 reservoir is established early during primary infection and is remarkably quantitatively and qualitatively stable. *Siliciano et al. (2003)* found a 44-month half-life for latently infected cells capable of producing replication-competent virus in the QVOA. Similarly, HIV-1 DNA levels and genetic compositions are very stable in patients receiving long-term suppressive ART (*von Stockenstrom et al., 2015*; *Besson et al., 2014*; *Josefsson et al., 2013*; *Kearney et al., 2014*; *Günthard et al., 1999*; *Evering et al., 2012*; *Kieffer et al., 2004*). Early ART reduces the reservoir's size and genetic complexity (*Chomont et al., 2009*; *Josefsson et al., 2013*; *Lori et al., 1999*; *Strain et al., 2005*). Most studies suggest that the HIV-1 reservoir is maintained by the physiological homeostasis of memory CD4 cells that in part involves occasional expansion and contraction of individual CD4 cell clones (*von Stockenstrom et al., 2015*; *Chomont et al., 2011*, *2009*). However, the results of some studies have suggested that persistent virus replication may be an important contributor to the maintenance of the HIV-1 reservoir (*Buzón et al., 2010*; *Yukl et al., 2010*). Recently, *Lorenzo-Redondo et al. (2016)* reported evidence of rapid HIV-1 evolution in lymphoid tissue reservoirs.

Despite their significance for HIV-1 cure efforts, relatively little is known about the pre-ART establishment and turnover of the HIV-1 reservoir. In this study, we characterized the establishment and maintenance of the HIV-1 DNA reservoirs in 10 patients. We previously studied the evolution of HIV-1 in these patients before ART by whole genome deep-sequencing of HIV-1 RNA in longitudinal plasma samples (*Zanini et al., 2015*). We now sequenced HIV-1 DNA from peripheral blood mononuclear cells (PBMCs) from these patients after many years of suppressive ART and compared these reservoir DNA sequences with the replicating HIV populations that were present before ART. The collection dates of all available samples relative to start of treatment are presented in *Figure 1*.

We found that the HIV-1 DNA populations remained genetically stable for up to 18 years after the start of suppressive ART. The absence of genetic changes indicates that viral evolution and replication are not important mechanisms for the maintenance of HIV-1 reservoirs during supressive ART. We also found that the variants that were replicating shortly before start of ART were overrepresented in the HIV-1 DNA reservoirs. This excess of late variants in the DNA reservoirs indicated that proviral HIV-1 variants continued to turn over with a half-life of approximately one year until the patients began therapy. ART effectively froze the composition of the HIV-1 DNA reservoir in the state it had at start of therapy.

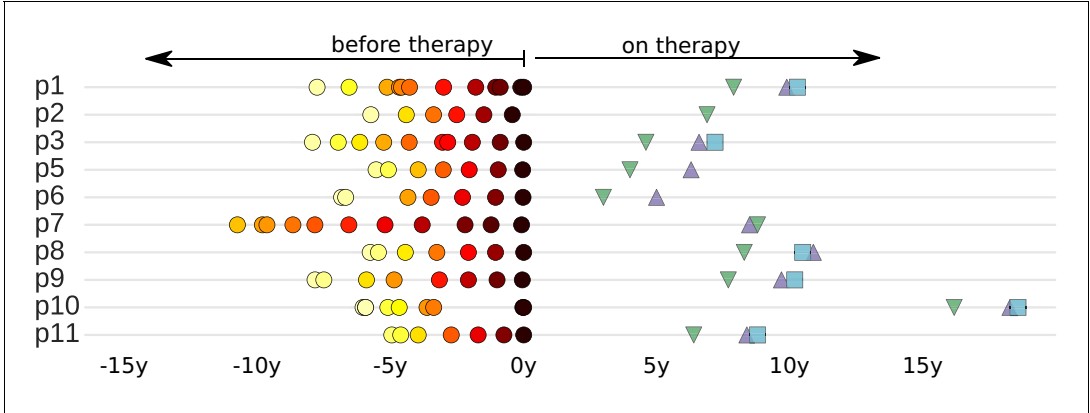

**Figure 1.** Sampling times before and after start of suppressive antiretroviral therapy. For each study participant, the thick grey bar indicates the period of untreated HIV-1 replication. Circles indicate the collection times of the plasma samples used for whole genome deep sequencing of the HIV-1 RNA populations (*Zanini et al., 2015*). Triangles and squares indicate the collection times of the PBMC samples used for p17gag deep sequencing of the HIV-1 DNA populations. All times are relative to start of therapy.

# Results

## Patients and samples

The study population consisted of 10 HIV-1 infected patients who were diagnosed in Sweden between 1990 and 2003. The following criteria were used to select the patients: (1) A relatively well-defined time of infection; (2) Being treatment-naive for $\geq$5 years; and (3) Receiving suppressive ART (plasma HIV-1 RNA levels continuously <50 copies/ml) for $\geq$2 years. We previously characterized HIV-1 RNA populations in longitudinal plasma samples (6–12 per patients over 5–8 years). Results for 9 out of the 10 patients were reported in *Zanini et al. (2015)*. The tenth patient (p7) was part of the previous study, but was not included in the final analyses of that study because the sequencing of plasma samples obtained during the first years after infection had failed. The results for the patient characteristics are presented in *Figure 1* and *Table 1*.

For this study, we obtained sequence data from the HIV-1 DNA in viral reservoirs using deep sequencing of the p17gag region of the HIV-1 genome in DNA prepared from PBMCs. Longitudinal PBMC samples (1–3 samples per patient for up to a 2.6-year period) were obtained 3–18 years after the start of suppressive ART (*Figure 1* and *Table 1*). We defined viral DNA reservoirs as HIV-1 p17gag sequences that were still present in PBMCs after at least 2 years of suppressive ART. HIV-1 DNA template numbers were quantified by limiting dilution by the same p17gag PCR that was used for sequencing. Identical p17gag sequences were merged into haplotypes while preserving their abundance. Minor haplotypes were merged with major haplotypes if they differed by one mutation (see Materials and methods section). Processed sequence data are available at hiv.tuebingen.mpg.de. Raw sequencing reads from all HIV-1 DNA samples were deposited in the European Nucleotide archive (study accession number PRJEB13841; sample accession numbers ERS1138001-ERS1138025).

## Proviral DNA sequences reflect pretreatment RNA sequences

The HIV-1 DNA sequences recapitulate the diversity observed in RNA sequences before treatment, often with exact sequence matches, *Figure 2* and *Figure 2—figure supplement 1*. While we observed large variations in the abundance of haplotypes with sequence read frequencies varying between 0.1 and 50% (*Figure 2—figure supplement 2*), the high similarities between RNA and DNA sequences confirmed that our characterization of proviral diversity was specific and sensitive. Variation in haplotype abundance likely reflects clonal expansion (*Josefsson et al., 2013*; *von Stockenstrom et al., 2015*), independent integration of identical sequences, and resampling of

**Table 1.** Summary of patient characteristics.

| Patient | Gender | Transmission | Subtype | Age[*] | HIV RNA from plasma | | HIV DNA from PBMCs | |
|---------|--------|--------------|---------|--------|-----------|----------------------|------------------------|-------------|
| | | | | | # samples | First/last since EDI[†] | Time on ART[†] | # templates |
| p1 | F | HET | 01_AE | 37 | 12 | 0.3 | 8.2 | 7.9/9.9/10.4 | 820/148/38 |
| p2 | M | MSM | B | 32 | 6 | 0.2 | 5.5 | 6.9 | 75 |
| p3 | M | MSM | B | 52 | 10 | 0.4 | 8.4 | 4.6/6.7/7.2 | 243/102/108 |
| p5 | M | MSM | B | 38 | 7 | 0.4 | 5.9 | 4.0/6.3 | 180/72 |
| p6 | M | HET | C | 31 | 7 | 0.2 | 7.0 | 3.0/5.0/5.5 | 115/15/nd |
| p7 | M | MSM | B | 31 | 11 | 6.3[‡] | 16.1 | 6.3/8.4/8.8 | 88/279/108 |
| p8 | M | MSM | B | 35 | 7 | 0.2 | 6.0 | 8.4/10.6/10.9 | 180/55/175 |
| p9 | M | MSM | B | 32 | 8 | 0.3 | 8.1 | 7.7/9.7/10.2 | 60/72/72 |
| p10 | M | MSM | B | 34 | 9 | 0.1 | 6.2 | 16.2/18.3/18.6 | 249/116/51 |
| p11 | M | MSM | B | 53 | 7 | 0.6 | 5.6 | 6.4/8.4/8.8 | 124/120/123 |

[*]at diagnosis;

[†]EDI: estimated date of infection; all times are given in years;

[‡]sequencing failed in earlier samples due to low plasma HIV-1 RNA levels.

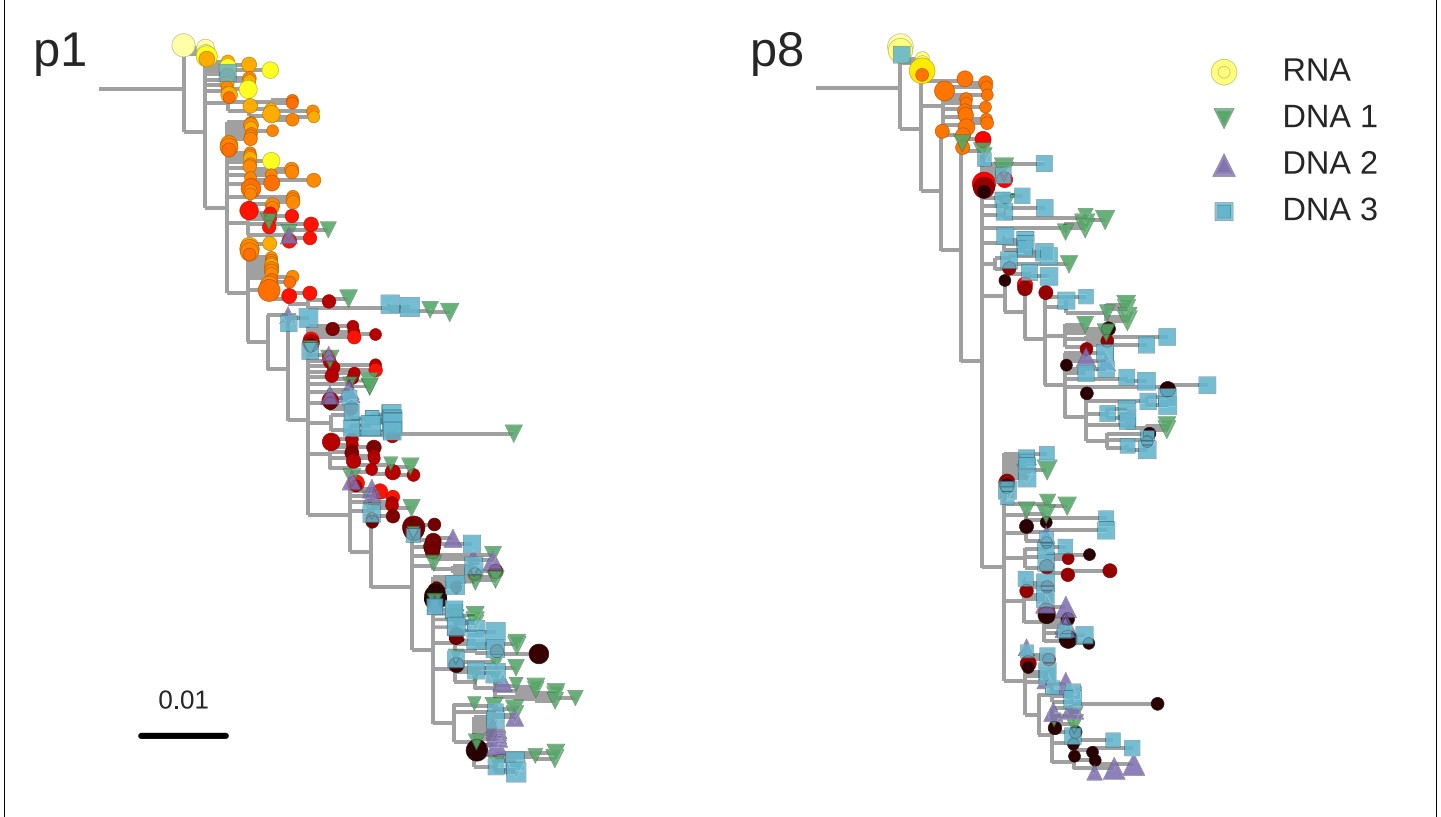

**Figure 2.** Reconstructed phylogenetic trees of plasma HIV-1 RNA sequences (circles) and PBMC HIV-1 DNA (triangles and squares) from two patients. The RNA sequences were obtained from plasma samples collected before the start of suppressive antiretroviral therapy (ART). DNA sequences 1, 2, and 3 were obtained from PBMCs collected after many years of suppressive ART (see *Figure 1*). The symbol colors indicate the sample date relative to the start of therapy and use the same color scale as in *Figure 1*. The symbol size indicates the fraction of reads represented by the node. The trees were built using the FastTree software (see Materials and methods section) (*Price et al., 2010*). Analogous phylogenetic trees for the remaining eight study patients are presented in *Figure 2—figure supplement 1*.

The following figure supplements are available for figure 2:

**Figure supplement 1.** Phylogenetic trees of plasma HIV-1 RNA and PBMC HIV-1 DNA sequences from all patients included in the study.

**Figure supplement 2.** Distributions of frequencies of haplotypes.

**Figure supplement 3.** Distributions of mutations in reads classified as hypermutanted or as non-hypermutated.

the same original DNA templates during sequencing. However, the specific contributions of these distinct mechanisms were difficult to determine in our sequencing results.

The estimated numbers of HIV DNA templates, the numbers of distinct haplotypes observed, and the fractions of haplotypes present in multiple samples are presented in *Supplementary file 1*. If a haplotype was present at a frequency >1%, it was also present in another sample from the same patient in approximately one-third (median 0.29) of all cases.

## Hypermutated sequences are frequent in HIV-1 reservoirs

We found that substantial proportions (median 14%; range 0–42%) of the p17gag DNA sequences from the viral reservoirs were hypermutated and were therefore expected to be replication incompetent (*Figure 2—figure supplement 3*). This result was consistent with the results of earlier research, which showed that 9–30% of sequences were hypermutated (*Josefsson et al., 2013*; *Bruner et al., 2015*; *Kieffer et al., 2005*). A small proportion of sequences had stop codons that were not obviously due to G-to-A hypermutation (mean 3%, range 0–12%). A proportion of sequences without

overt inactivating mutations were likely also replication incompetent due to mutations or deletions outside of p17gag (*Ho et al., 2013*; *Bruner et al., 2016*). We excluded hypermutated sequences from the main analyses, but we also performed complementary analyses that included hypermutated sequences.

## Lack of evidence of persistent replication in HIV-1 DNA reservoirs

Whether or not HIV-1 reservoirs are maintained by persistent replication remains controversial (*von Stockenstrom et al., 2015*; *Chomont et al., 2011*, *2009*; *Buzón et al., 2010*; *Yukl et al., 2010*; *Lorenzo-Redondo et al., 2016*; *Evering et al., 2012*). We used the p17gag DNA sequences from viral reservoirs to search for evidence of sequence evolution, which should occur if the virus was replicating. Root-to-tip distances for plasma RNA populations and PBMC DNA populations were calculated relative to the major RNA haplotype present in the first plasma sample.

*Figure 3* presents the results for temporal changes in root-to-tip distances in HIV-1 RNA and DNA populations obtained before and after the start of suppressive ART, respectively. Plasma HIV-1 RNA populations obtained before the start of ART evolved at a relatively constant rate (*Zanini et al., 2015*); there was a steady increase in mean root-to-tip distances over time in *Figure 3*. In sharp contrast, the HIV-1 DNA populations obtained after 3–18 years of suppressive therapy had stable root-to-tip distances. Hypermutated DNA sequences had larger root-to-tip distances, but these distances were also stable over time (*Figure 3—figure supplement 1*). To rule out biases due to clonal expansion or PCR resampling, or both, we repeated this analyses while counting each unique sequence only once; we obtained essentially the same results (*Figure 3—figure supplement 2*).

The results for the rates of evolution before and after the start of suppressive ART are presented in *Table 2*. In all 10 patients we found statistically significant evolution of plasma RNA sequences before the start of therapy, with rates $1-4 \times 10^{-3}$/year. In contrast, DNA sequences showed no signal of statistically significant evolution in DNA reservoirs during suppressive ART.

Taken together, our results did not indicate that persistent HIV-1 replication acts as a mechanism to maintain the HIV-1 reservoir during suppressive therapy.

## Time of deposition of reservoir HIV-1 DNA sequences

The phylogenetic analyses indicated that most of the HIV-1 DNA variants present in the PBMCs matched the HIV-1 RNA variants detected in the plasma samples obtained shortly before the start of therapy (*Figure 2* and *Figure 2—figure supplement 1*). However, the phylogenetic trees also revealed that DNA variants that matched the earliest plasma variants were present in some patients.

To investigate when the PBMC HIV-1 DNA variants were deposited in the viral reservoirs, we compared the on-treatment PBMC DNA sequences with the longitudinal pre-treatment plasma RNA sequences. For each p17gag DNA sequence, we determined the pre-treatment plasma sample and the RNA haplotype that was the most likely source. This method assigned most of the HIV-1 DNA sequences to the plasma samples closest to the start of treatment (*Figure 4*, panel A and *Figure 4—figure supplement 1*). A combined analysis of the data for all patients (*Figure 4*, panel C) indicated that approximately 60% of the DNA reads were most similar to RNA variants that were present in plasma samples obtained during the last year before the start of therapy. The representation of the variants present in the earlier plasma samples decreased; the half-life was 0.75 years going backward in time from the start of therapy. Analyses in which each unique sequence was counted only once revealed very similar results, which indicated that the findings were robust to possible sampling biases from clonal expansions or PCR resampling, or both (*Figure 4—figure supplement 2*).

Sequences that matched sequences at earlier plasma sampling time points were found as minor variants among the p17gag DNA sequences (*Figure 4*, panel B). Among these minor variants, the DNA sequences matching plasma variants obtained within six months post-infection were overrepresented in some patients (14%, 2.4%, 42%, <1%, and 6.9% of all the reads in patients 2, 3, 6, 8, and 11, respectively). On average approximately 5% of the reads matched plasma variants from the first 6 months after the estimated date of infection (EDI) (*Figure 4*, panel B). If the outlying data for patient six were omitted, this fraction was 2%. Even 2% was unexpectedly high given the rapid DNA decay during untreated HIV-1 infection. If the rates of seeding into the reservoirs were stable over time, we would have expected a mean value of 0.14% (range < 0.001–0.32% across patients) of the

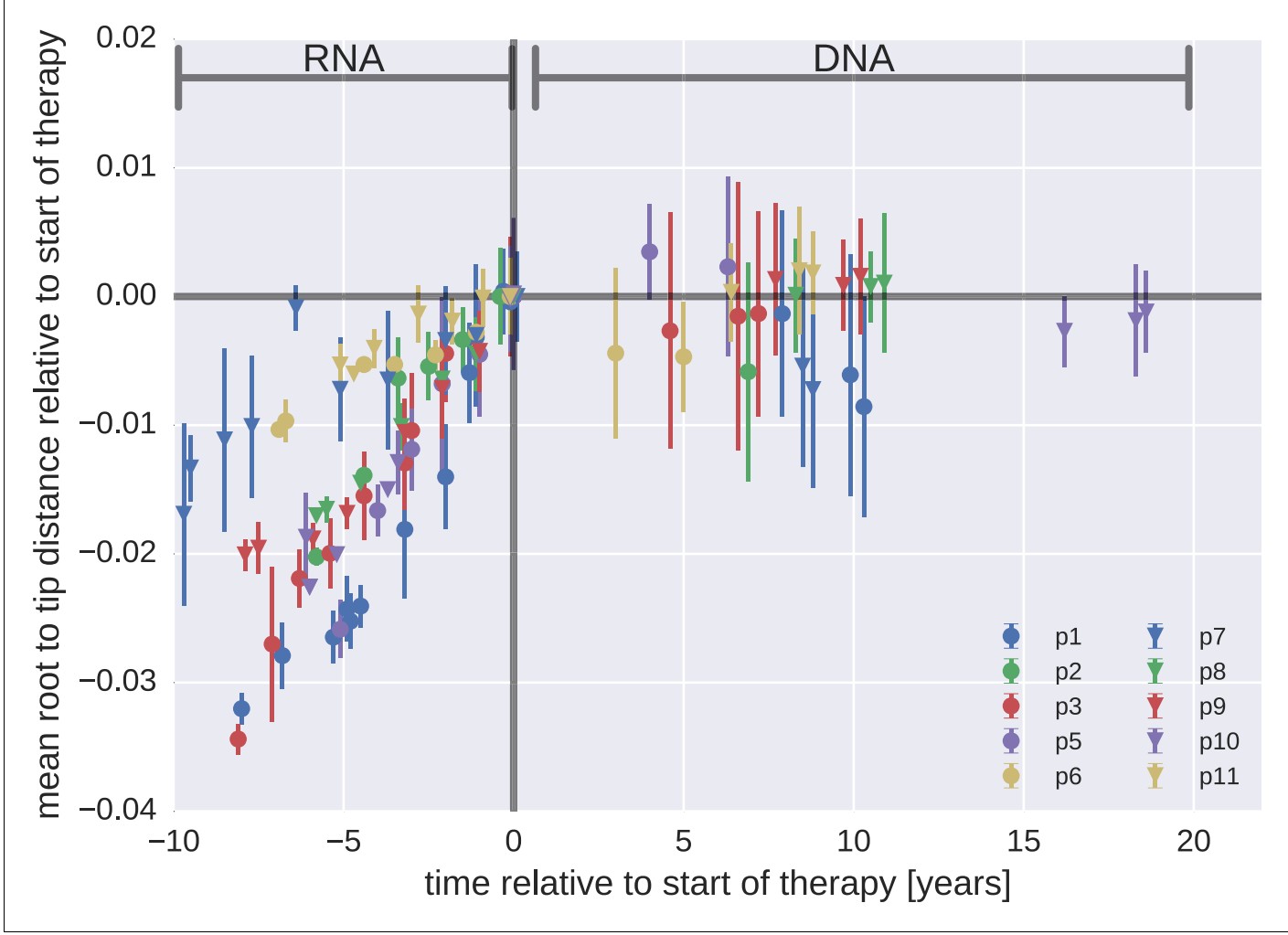

**Figure 3.** Root-to-tip distances. The plasma HIV-1 RNA sequences evolved steadily before the start of antiretroviral therapy (ART), while no evidence of evolution was found in the PBMC HIV-1 DNA sequences obtained after the start of ART. For each patient, we have samples obtained before or at the start of therapy (HIV-1 RNA from plasma), and samples obtained after the start of therapy (HIV-1 DNA from PBMCs). The error bars indicate ±± one standard deviation of the root-to-tip distances. The figure contains data on DNA sequences not classified as hypermutants. The analogous figures for the hypermutants and haplotypes are presented in *Figure 3—figure supplements 1* and *2*, respectively.

The following figure supplements are available for figure 3:

**Figure supplement 1.** Mean root-to-tip distances for plasma HIV-1 RNA sequences obtained before the start of ART and PBMC HIV-1 DNA sequences obtained after the start of ART.

**Figure supplement 2.** Mean root-to-tip distances for plasma HIV-1 RNA sequences obtained before the start of antiretroviral therapy (ART) and PBMC HIV-1 DNA sequences obtained after the start of ART.

**Figure supplement 3.** Short lived cells can generate a false signal of evolution.

DNA sequences to match the earliest plasma variants. This result suggested that massive seeding of HIV DNA into the viral reservoirs occurred during the first weeks and months post-infection.

In summary, viral variants that replicated shortly before the start of suppressive ART were over-represented in the HIV-1 DNA reservoirs. This result indicated that the infected cells were turning over (approximately one-year half-life) for as long as the patients were untreated. Suppressive ART halted this turnover.

**Table 2.** Rates of evolution in plasma HIV-1 RNA and PBMC HIV-1 DNA sequences obtained before the start and after the start of suppressive antiretroviral therapy, respectively.

| Patient | RNA rate | | DNA rate | |
|---|---|---|---|---|
| | [Year$^{-1}$] | p-value | [Year$^{-1}$] | p-value |
| p1 | $4.4 \times 10^{-3}$ | $<10^{-6}$ | $-6 \times 10^{-4}$ | 0.22 |
| p2 | $3.7 \times 10^{-3}$ | $<10^{-2}$ | $-8 \times 10^{-4}$ | – |
| p3 | $4.1 \times 10^{-3}$ | $<10^{-6}$ | $-2 \times 10^{-4}$ | 0.39 |
| p5 | $4.8 \times 10^{-3}$ | $<10^{-3}$ | $4 \times 10^{-4}$ | 0.45 |
| p6 | $1.4 \times 10^{-3}$ | $<10^{-3}$ | $-9 \times 10^{-4}$ | 0.22 |
| p7 | $1.3 \times 10^{-3}$ | $<10^{-2}$ | $-7 \times 10^{-4}$ | 0.14 |
| p8 | $2.9 \times 10^{-3}$ | $<10^{-5}$ | $8 \times 10^{-5}$ | 0.22 |
| p9 | $2.6 \times 10^{-3}$ | $<10^{-4}$ | $1 \times 10^{-4}$ | 0.12 |
| p10 | $3.6 \times 10^{-3}$ | $<10^{-5}$ | $-1 \times 10^{-4}$ | 0.20 |
| p11 | $1.2 \times 10^{-3}$ | $<10^{-2}$ | $2 \times 10^{-4}$ | 0.16 |

## Discussion

We investigated the composition and turnover of HIV-1 DNA sequences in viral reservoirs in patients receiving long-term suppressive therapy. The reservoir HIV-1 DNA populations were remarkably stable and showed no evidence of active replication during ART. Since we had previously characterized the HIV-1 populations in these patients prior to ART, we could determined how sequences in the HIV-1 reservoir related to pre-treatment populations. In particular, we were able to show that sequences in HIV-1 DNA reservoirs mainly derived from the populations present during the last year before the start of suppressive therapy.

Our results indicate that persistent HIV-1 replication is not a mechanism for maintenance of HIV-1 reservoirs during suppressive therapy (*Figure 3* and *Table 2*). This conclusion differs from the results of a recent report by *Lorenzo-Redondo et al. (2016)* and of a few earlier reports (*Yukl et al., 2010*; *Buzón et al., 2010*). However, our results are consistent with the results of other earlier studies (*von Stockenstrom et al., 2015*; *Besson et al., 2014*; *Josefsson et al., 2013*; *Kearney et al., 2014*; *Günthard et al., 1999*; *Evering et al., 2012*; *Kieffer et al., 2004*). *Lorenzo-Redondo et al. (2016)* compared genetic diversity in HIV-1 RNA in plasma samples at the start of therapy with HIV-1 DNA sequences obtained from blood and tissues at baseline, 3 months, and 6 months after the start of treatment. The authors reported very high rates of evolution ($7.4-12 \times 10^{-3}$ changes per site per year); these rates are approximately 3- to 5-fold greater than those typically observed in gag and pol of replicating RNA populations. Such rapid evolution is incompatible with the lack of observable changes in the reservoir sequences over the 20 times longer time intervals reported here (*Figure 3—figure supplement 2*).

Without longitudinal data on the evolution of HIV-1 populations prior to treatment, the nature of the changes reported by *Lorenzo-Redondo et al. (2016)* are difficult to determine. It is possible that the reported temporal signal arises from changes in DNA reservoir composition during the first 6 months of therapy when short-lived cells infected with recently replicating virus variants gradually disappear. Death of short-lived cells increases the proportions of longer-lived cells that sample further back into the history of the infection. This scenario (*Figure 3—figure supplement 3*) would result in sequence changes that do not indicate (forward) evolution. Instead these changes can create an impression of 'backward' evolution towards older HIV variants due the preferential pruning of later variants.

*Lorenzo-Redondo et al. (2016)* investigated HIV-1 DNA sequences in tissue and PBMC samples, but we only examined PBMC samples. However, Lorenzo-Redondo et al. found similar rates in PBMC compared to tissue samples. Furthermore, tissue and blood HIV-1 DNA variants should be well-mixed during the time period that we investigated (*Josefsson et al., 2013*; *von Stockenstrom et al., 2015*; *Lorenzo-Redondo et al., 2016*).

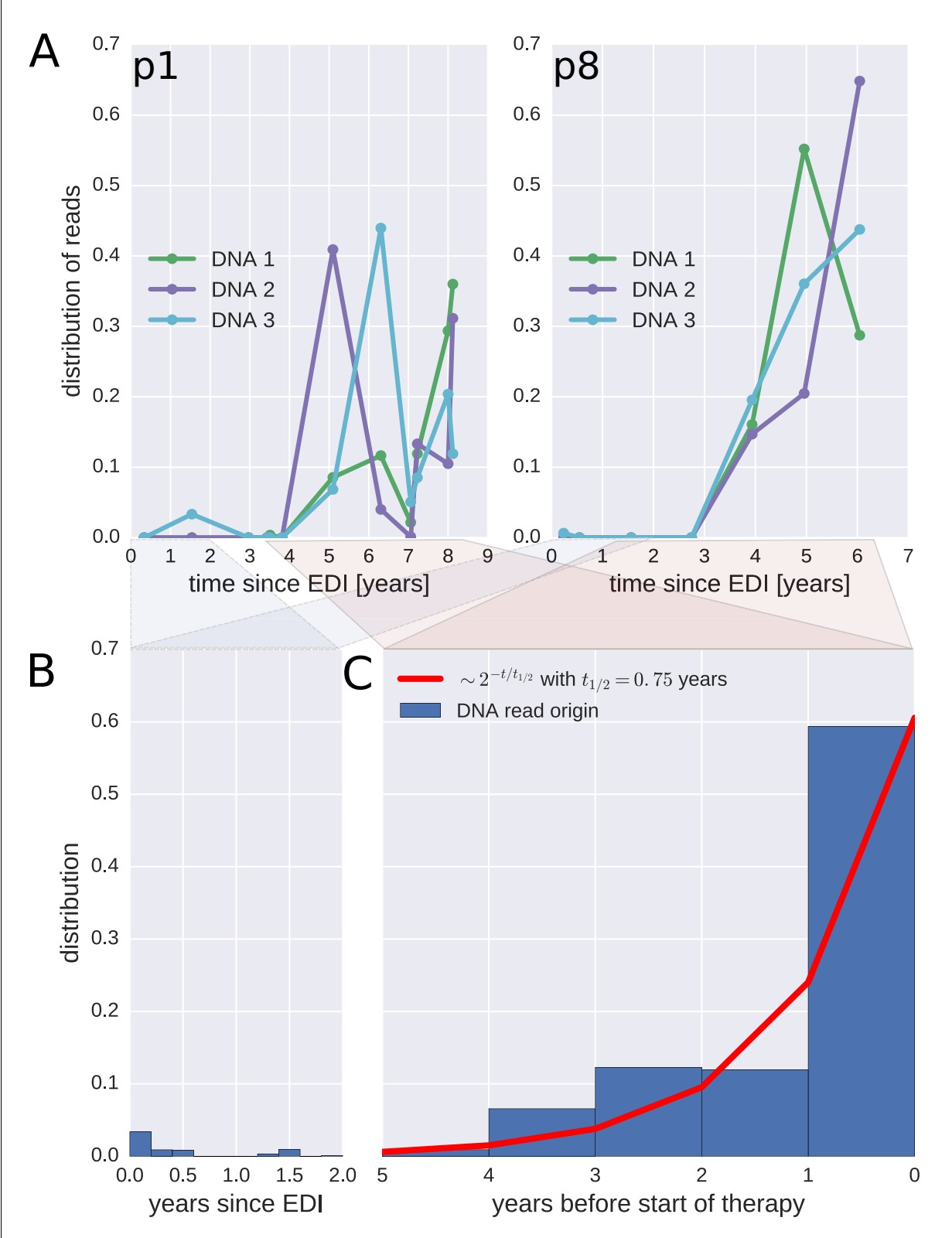

**Figure 4.** Probable origin of sequences in the DNA reservoir. For each HIV-1 DNA read obtained from the PBMCs, we determined the pre-treatment plasma sample and HIV-1 RNA variant that was the most likely origin of the read. Panel A presents the results for the distribution of these pre-treatment assignments for HIV-1 DNA reads from the three PBMC samples (DNA 1, 2, and 3) of patients p1 and p8 (compare trees in *Figure 2*). The analogous graphs for all patients are presented in *Figure 4—figure supplement 1*. A large fraction of the reservoir DNA sequences was most closely related to

*Figure 4 continued on next page*

*Figure 4 continued*

the RNA sequences present in the plasma samples obtained during the last year before the start of therapy. Panels B and C present summaries of the results for the distributions of the likely origins of the reservoir reads for all samples as a function of time since estimated date of infection (EDI) and time before treatment start, respectively. The distribution of reads decays with the time from treatment start (half-life of approximately 0.75 years) (Panel C). A small fraction of reads was estimated to originate from plasma samples obtained shortly post-infection (Panel B). We found these early reservoir sequences in 50% of the patients.

The following figure supplements are available for figure 4:

**Figure supplement 1.** The distribution of plausible seeding times of reservoir HIV-1 DNA sequences for all 10 patients.

**Figure supplement 2.** Same as *Figure 4—figure supplement 1*, but counting each sequence once instead of weighted by the number of reads it represents.

Both *Lorenzo-Redondo et al. (2016)* and we studied DNA sequences in HIV-1 reservoirs, which contain high proportions of defective virus (*Ho et al., 2013*; *Bruner et al., 2016*). The absence of provirus evolution or turnover does not fully exclude the possibility that there was replication and evolution of replication-competent viruses present in the reservoirs. However, if such replication occurred, it involved a minority of infected cells and occurred below the detection limit of the deep-sequencing method. To enrich for replication-competent and putatively evolving virus, QVOA followed by sequencing of virus released into the supernatant should be performed, not sequencing the total HIV-1 DNA (as done by *Lorenzo-Redondo et al. [2016]*, us, and others [*Josefsson et al., 2013*; *Kieffer et al., 2005*; *Evering et al., 2012*]). Our finding of genetic stability in the DNA reservoirs is consistent with *Josefsson et al. (2013)* and *von Stockenstrom et al. (2015)*. They found that defective HIV-1 DNA integrants present during long-term effective ART appear to be maintained by proliferation and longevity of infected cells and not by ongoing viral replication.

We had detailed longitudinal data on the evolution of the plasma HIV-1 RNA population from the time of infection to the start of suppressive ART. Therefore, we could determine the time at which the viruses in the DNA reservoirs had been deposited (*Figure 4* and supplements). We found that a majority of the variants present in the HIV-1 DNA reservoirs were derived from HIV-1 RNA variants that had replicated during the last year before the start of suppressive ART. In contrast, *Frenkel et al. (2003)* reported the persistence of greater numbers of earlier versus more recent virus variants in a few children receiving suppressive ART. We found a similar, but much less prominent, presence of variants replicating during the first 6 months post-infection (*Figure 4*, panel B). These early variants were overrepresented relative to the expected level if deposition had been uniform over time. This overrepresentation could be due to the high viral load present during primary HIV-1 infection and is consistent with results by *Bruner et al. (2016)*, who found that defective proviruses rapidly accumulate during acute HIV-1 infection. However, we observed such an excess of early variants in only 5 out 10 patients.

Defective HIV-1 proviruses are unique in vivo labels of individual memory CD4 cell clones. Similar to the sequencing of T-cell receptors, these labels can be used to track the fates of the memory CD4 cells (*Robins, 2013*). This strategy was used by *Imamichi et al. (2014)*, who found that a T-cell clone carrying a defective HIV-1 provirus can persist for >17 years. Similarly, prenatally formed T-cell receptors shared by twins have lifetimes that are >30 years (*Pogorelyy et al., 2016*). Our results are consistent with such long T-cell life times once HIV replication is suppressed by ART. In untreated infection, however, we estimate much faster turnover of infected cells with a half-life of approximately one year. This conclusion is based on the observation that most HIV-1 DNA sequences derive from replicating virus shortly before the start of ART. The results of earlier studies, which were based on different types of CD4 cell labelling, have indicated that CD4 cell die at a 3- to 4-fold increased rate in untreated HIV-1-infected patients, compared with uninfected individuals and with patients receiving suppressive ART (*Hellerstein et al., 1999*; *McCune et al., 2000*; *Ribeiro et al., 2002*). The more dramatic difference that we observed might be due to that earlier studies estimated lifespans of individual cells, whereas we estimated the lifespans of CD4 cell clones carrying defective proviruses (i.e., infected cells and their daughter cells).

Our study had several limitations. We did not sort cells and therefore could not investigate whether there were differences in HIV-1 turnover rates between different types and subsets of cells (e.g., memory CD4 cells and their subsets). However, it is reasonable to assume that most of our HIV-1 DNA sequences were from memory CD4+ T-lymphocytes because others have found that these cells are the main HIV-1 reservoir (*Eriksson et al., 2013*; *Chun et al., 1997*, *1995*). Because we sequenced a relatively short region of the HIV-1 genome, we could not reliably distinguish between replication-competent and defective viruses. We did not find evidence of evolution in proviral DNA sequences, but we cannot rule out the possibility that a small subset of viruses was replicating and remained undetected among the many defective viruses. However, our analyses of the composition and turnover of the virus reservoir were aided by the fact that most of the viruses were defective because these proviruses acted as inert in vivo labels of CD4 cell clones. We observed large variations in the abundance of sequence haplotypes that likely indicate the presence of clonal expansions (*Josefsson et al., 2013*; *von Stockenstrom et al., 2015*), independent integrations of identical sequences, and resampling of the same original DNA templates during sequencing. Given our sequencing method, we could not determine the relative contributions of these mechanisms. We are using Primer ID sequencing (*Jabara et al., 2011*) to better understand the in vivo dynamics of different viral haplotypes. Analysis of HIV integration sites represents an alternative method of clonal expansion identification (*Maldarelli et al., 2014*; *Cohn et al., 2015*). However, such analysis was not possible for us because we did not sequence the extreme ends of the LTRs. We analyzed p17gag because it was part of our plasma HIV RNA dataset and because it has sufficient genetic diversity to allow for accurate and valid comparisons between plasma RNA and PBMC DNA HIV populations. Results from several analyses suggested that clonal expansion did not affect our results in a qualitative manner. Clonal expansion would have resulted in artefactual over-representation of some sequences. To control for this, we repeated the analyses in *Figures 3* and *4* and counted each haplotype only once. We obtained very similar results (*Figure 3—figure supplement 2* and *Figure 4—figure supplement 2*). Consistent with our results and interpretation, *Cohn et al. (2015)* found only a limited decrease in single integrations following the start of ART (from approximately 70 to 50%).

In summary, we found compelling evidence against persistent viral replication as a mechanism to maintain the latent HIV-1 DNA reservoir during suppressive therapy. We also found that most of the latently infected cells present during long-term suppressive ART were infected shortly before the start of ART, and that T-cell turnover slowed down dramatically when ART began.

## Materials and methods

### Ethical statement
The study was performed according to Declaration of Helsinki requirements. Ethical approval was granted by the Regional Ethical Review Board, Stockholm, Sweden (Dnr 2012/505 and 2014/646). Written and oral informed consent was obtained from each patient that participated in the study.

### Patients
The study population consisted of 10 HIV-1-infected patients who were diagnosed in Sweden between 1990 and 2003. These patients had been included in a recent study of the population genomics of intrapatient HIV-1 evolution (*Zanini et al., 2015*). The patients were selected based on the following inclusion criteria: (1) A relatively well-defined time of infection (based on a negative HIV antibody test <2 years before a first positive test or a laboratory-documented primary HIV infection); (2) No ART during a minimum period of approximately 5 years after diagnosis; (3) Availability of biobank plasma samples covering this time period; and (4) Later initiation of successful ART (plasma viral levels <50 copies/ml) for ≥2 years. As previously described, 6–12 plasma samples per patient were retrieved from biobanks and used for full-genome HIV-1 RNA sequencing (*Zanini et al., 2015*). The same patient nomenclature was used in both studies. For this study, the same 10 patients were asked to donate 70 ml fresh EDTA-treated blood on up to three occasions during a 2.5-year period. These blood samples were obtained 3–18 years after the start of successful ART. The clinical and laboratory findings (including Fiebig staging and BED testing) were used to calculate the EDI, as

previously described (*Zanini et al., 2015*). The results for the characteristics of the patients and the samples are presented in *Table 1*.

## HIV-1 RNA sequencing from plasma

Whole-genome deep-sequencing of the virus RNA populations present in the plasma samples obtained before the start of therapy was performed as previously described (*Zanini et al., 2015*). Briefly, the total plasma RNA was extracted using an RNeasy Lipid Tissue Mini Kit (Qiagen Cat No. 74804). The extracted RNA was amplified using a one-step RT-PCR, outer primers for six overlapping regions, and Superscript III One-Step RT-PCR with Platinum Taq High Fidelity High Enzyme Mix (Invitrogen, Carlsbad, California, US). An optimized Illumina Nextera XT library preparation protocol and a kit from the same supplier were used to build the DNA libraries, which were sequenced using an Illumina MiSeq instrument and 2 × 250 bp or 2 × 300 bp sequencing kits (MS-102–2003/MS-10–3003). For the present study, a section of the p17gag region of the HIV-1 genome was extracted from the full-genome RNA data set. A median number of 146 (inter-quartile range 56–400) high quality reads was obtained. The cDNA template numbers are available in *Zanini et al. (2015)*.

## HIV-1 DNA sequencing from PBMCs

A total of approximately 70 ml fresh whole blood was split between seven Vacutainer (EDTA) tubes. The PMBCs were isolated using Ficoll-Paque PLUS (GE Healthcare Bio-Sciences AB, Uppsala, Sweden) centrifugation according to the manufacturer's instructions. Total DNA was extracted from the PBMCs using the OMEGA E.Z.N.A Blood DNA Mini Kit (Omega Bio-Tek, Norcross, Georgia) or the QIAamp DNA Blood Mini Kit (Qiagen GmbH, Hilden, Germany) according to the manufacturer's instructions. The amount of DNA was measured using a Qubit dsDNA HS Assay Kit (Invitrogen, Eugene, Oregon, USA). Patient-specific nested primers (Integrated DNA Technologies) were used to amplify a 387 bp long portion of the p17gag gene corresponding to positions 787 to 1173 in the HxB2 reference sequence. The primers were designed based on the plasma RNA sequences from each patient (*Supplementary file 2*). Outer primers and Platinum Taq DNA Polymerase High Fidelity (Invitrogen, Carlsbad, California, US) were used for the first PCR. The protocol began with a denaturation step at 94°C for 2 min followed by 15 PCR cycles of denaturation at 94°C for 20 s, annealing at 50°C for 20 s, extension at 72°C for 30 s, and a final extension step at 72°C for 6 min. For the second PCR, 2.5 µl of the product from the first PCR was amplified using inner primers and the cycle profile and enzyme as for the first PCR. Amplified DNA was purified using Agencourt AMPure XP (Beckman Coulter, Beverly, Massachusetts) and quantified using Qubit. For each sample, the number of HIV-1 DNA templates used for sequencing was roughly quantified in triplicate by limiting dilution using the same PCR conditions, three dilutions (usually 0.5 µg, 0.1 µg, 0.02 µg DNA), and Poisson statistics. The plasmids NL4-3 and SF162 were used for control experiments, which were performed to evaluate PCR-induced recombination. Equal proportions of the plasmids were spiked into human DNA and were amplified using the same PCR conditions as previously described. The results revealed that there was minimal PCR-induced recombination in this short amplicon.

## Sequencing and read processing

The HIV-specific primers were flanked by NexteraXT adapters. To construct sequencing libraries, indices and sequencing primers were added in 12–15 cycles of additional PCR. The amplicons were sequenced using an Illumina MiSeq instrument and 2 × 250 cycle kits. A total of 6500 to 190,000 (median 35,000) paired-end reads were generated per sample. The overlapping paired-end sequencing reads were merged to create synthetic reads spanning the entire p17gag amplicon. The nucleotide on the read with the higher quality score was used for the cases of disagreement between paired reads. We counted the numbers of times a particular p17gag sequence was observed, and performed subsequent analysis using read-abundance pairs. To reduce the effects of sequencing and PCR errors, we combined rare sequences (below frequency 0.002) with common sequences if they differed at no more than one position. Specifically, starting with the rarest sequences, we merged rare sequences with the most common sequence that was one base away. The cutoff value of 0.002 is the typical error frequency of the pipeline (*Zanini et al., 2015*). Less than 1 in 1000 reads that began with the HIV specific primer sequence could not be assigned with confidence to the previously determined RNA sequences. Most of these reads mapped to the human genome.

All analyses were performed using Python and the libraries NumPy, BioPython, and Matplotlib (*Cock et al., 2009*; *van der Walt et al., 2011*; *Hunter, 2007*). All read files have been uploaded to ENA with study accession number PRJEB13841.

## Hypermutation detection

To classify sequences into hypermutated and non-hypermutated sequences, we counted mutations at positions that did not vary in the RNA samples obtained before therapy. If greater than four mutations were observed, and at least one-half of them were G→A, the sequence was considered to be a hypermutant. The results for the distributions of the different transition mutations relative to the closest genome found in the RNA samples are presented in *Figure 2—figure supplement 3* for reads classified as hypermutants, or not. The results we obtained for sequences classified as non-hypermutants were very similar to the results obtained when only using sequences without stop codons.

## Phylogenetic analysis

We reconstructed phylogenetic trees using the approximate maximum likelihood method implemented in the FastTree software (*Price et al., 2010*). The tips were annotated with frequency, source, and sample date using custom-made Python scripts.

## Statistical analysis

The root-to-tip distances were calculated as the mean distance between a sample and the founder sequence, approximated by the consensus sequence of the first RNA sample. This root-to-tip distance was regressed against time to determine the rate of evolution in the absence of treatment. The root-to-tip sequence of the last RNA sample and the DNA samples was regressed against time to determine the rate of evolution on treatment. To determine the most likely seeding time for a PBMC p17gag DNA sequence, we calculated the likelihood of sampling this sequence given the SNP frequencies in each plasma RNA sample and assigned the sequence to the sample that had the greatest likelihood value.

Scripts used for processing and analysis of the data as well as plotting the results are available at https://github.com/neherlab/HIVEVO_reservoir.

## Acknowledgements

We would like to express our gratitude to the study participants.

## Additional information

### Competing interests

RAN: Reviewing editor, *eLife*. The other authors declare that no competing interests exist.

### Funding

| Funder | Grant reference number | Author |
|---|---|---|
| European Research Council | Stg. 260686 | Richard A Neher |
| Vetenskapsrådet | K2014-57X-09935 | Jan Albert |

The funders had no role in study design, data collection and interpretation, or the decision to submit the work for publication.

### Author contributions

JB, FZ, RAN, JA, Conception and design, Acquisition of data, Analysis and interpretation of data, Drafting or revising the article; LT, CL, GB, Acquisition of data, Analysis and interpretation of data, Drafting or revising the article

## Author ORCIDs

Fabio Zanini, http://orcid.org/0000-0001-7097-8539
Richard A Neher, http://orcid.org/0000-0003-2525-1407
Jan Albert, http://orcid.org/0000-0001-9020-0521

## Ethics

Human subjects: The study was conducted according to the Declaration of Helsinki. Ethical approval was granted by the Regional Ethical Review board in Stockholm, Sweden (Dnr 2012/505 and 2014/646). Patients participating in the study gave written and oral informed consent to participate.

## Additional files

### Supplementary files

• Supplementary file 1. Sequencing and hypermutation statistic for all samples. 'good' refers to proviral sequences that are not obviously defective, 'hyper' refers to those with an excess of G→A mutations. The column '% recaptured' indicates the fraction of sequences >1% frequency that were present in other samples from the same patient.

• Supplementary file 2. PCR primers used for HIV-1 p17gag amplification.

### Major datasets

The following dataset was generated:

| Author(s) | Year | Dataset title | Dataset URL | Database, license, and accessibility information |
|---|---|---|---|---|
| Brodin J, Zanini F, Thebo L, Lanz C, Bratt G, Neher RA, Albert J | 2016 | Establishment and stability of the latent HIV-1 DNA reservoir | http://www.ebi.ac.uk/ena/data/view/PRJEB13841 | Publicly available at the EBI European Nucleotide Archive (Accession no: PRJEB13841) |

The following previously published dataset was used:

| Author(s) | Year | Dataset title | Dataset URL | Database, license, and accessibility information |
|---|---|---|---|---|
| Zanini F, Neher R | 2015 | HIVEVO | http://www.ebi.ac.uk/ena/data/search?query=PRJEB9618 | Publicly available at the EBI European Nucleotide Archive (Accession no: PRJEB9618) |

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
