## [Decision Letter]

Thank you for submitting your article "Establishment and stability of the latent HIV-1 DNA reservoir" for consideration by *eLife*. Your article has been reviewed by three peer reviewers, and the evaluation has been overseen by Arup Chakraborty as the Senior Editor, and he also served as the Reviewing Editor. The reviewers have opted to remain anonymous.

The reviewers have discussed the reviews with one another and the Reviewing Editor has drafted this decision to help you prepare a revised submission.

Summary:

You have carried out a nice study of HIV evolution in patients prior to and after initiation of antiretroviral therapy. You show that the virus continues to evolve until ART is started. During ART, there is no further evolution. This conclusion is consistent with many previous studies. Although most studies of HIV evolution during ART have concluded that ART stops viral replication and thus halts evolution, a recent Nature paper by Wolinsky and colleagues provided some evidence for continuing evolution. Thus, the current paper is timely. The issue is of particular importance with regard to finding a cure for HIV investigation. Most investigators believe that the real barrier to cure is a stable latent reservoir. However, the finding that evolution is still occurring during ART would suggest that an additional problem is inadequate suppression of viral replication. The demonstration that evolution is halted strongly favors the idea that reservoirs are the barrier to cure, not inadequate ART. For these reasons, we think that the paper is likely to be accepted, provided the following comments are addressed.

1) You analyzed gag sequences in proviral DNA. However, it is not clear that most proviruses are profoundly defective. Even if the gag gene can be successfully amplified, there are likely to be defects elsewhere in the genome in many of the sequences. How does this affect the phylogenetic analysis? Is it appropriate to treat defective and replication-competent viral species the same way in such analyses? Some discussion of this issue would be helpful.

2) Paragraph 2 in Introduction (on QVOA) is difficult to understand. First, none of the QVOA information is used later in the text. The study quantifies neither the number of defective viruses nor the size of the reservoir. Second, while this is a matter of scholarship, the statement about PHA partial stimulation is problematic. There is no evidence that PHA in culture results in partial T-cell activation. The cited paper, Ho et al. 2013 only shows that there is partial HIV reactivation in culture, not the mechanism of how it arises. Partial reactivation of HIV may be due to intrinsic properties of the virus and should also be mentioned (Razooky et al. 2015, Rouzine et al. 2015).

3) A very strong part of the study is the comparison between HIV DNA post-ART and the HIV RNA pre-ART. However, there was not discussion or analysis of this. In our opinion, Figure 4 needs to be substantially 'fleshed out'. It is a little difficult to understand particularly panels A versus B. The low half-life of T cells before ART is noted, but then as in the subsection “Time for deposition of reservoir DNA sequences”, there were patients that showed 14% and 42% of their DNA sample matching the initial plasma RNA. Neither of these are small numbers given the half-lives (time period without ART shown in Figure 1) and given the previous panel showing most sequences came from just before ART. Can you explain this so that these apparently contradictory issues are resolved?

4) A more rigorous comparison to the data in Lorenzo-Redondo et al. (possibly in a new Discussion section) would be extremely helpful. The counter argument that will in all likelihood be used to rebut and limit this work as definitive is the classic "lack of data" argument (i.e., demonstrating that changes in the viral sequence occurred as a result of viral evolution is typically easier since there can be presence of signal in the sequence that appears to show such changes, as in Lorenzo-Redondo et al; in contrast, lack of data is always less conclusive because, for several reasons, this signal may have been missed, and the authors state this in the text. However, by comparison to the Lorenzo-Redondo study, the data here has more patients and time points but does not have multiple organs. It would be helpful if it is pointed out how much evolution was observed in Lorenzo-Redondo et al. PBMC data, in how many patients, and then show how unlikely or likely it would be for them to have not seen it. If Lorenzo-Redondo et al. see it in PBMC data and the results here do not, why would that be? Also, in the second paragraph of their Discussion, the authors point out that Lorenzo-Redondo et al. report a higher than normal rate of evolution. In that paper however, different work was cited claiming their rates were normal. I think the authors should expand on this and specifically point out references and rates in the presence and absence of ART.

5) Related to the above point, why does the analysis not account for clonal expansion? The authors should have done this analysis (see Maldarelli et al., 2014 or Cohn et al., 2015) to estimate the extent of expansion rather than leave that to speculation (even for a single patient where they assumed this was the top contribution). This would have taken out a major source of possible confusion. What if the sample prep or methodology are somehow dominated by clonal expansion which is why they don't capture any variation?

6) Figure 4: it is noted that most proviral sequences were assigned to a sample just before treatment. How many were assigned and how many could not be assigned by patient?

---

## [Author Response]

[…]

*1) You analyzed gag sequences in proviral DNA. However, it is not clear that most proviruses are profoundly defective. Even if the gag gene can be successfully amplified, there are likely to be defects elsewhere in the genome in many of the sequences. How does this affect the phylogenetic analysis? Is it appropriate to treat defective and replication-competent viral species the same way in such analyses? Some discussion of this issue would be helpful.*

It is true that most proviruses are likely to be defective. The dominating forms of inactivating genetic changes are large deletions and hypermutation. This has been fully explained in the Introduction (second paragraph) where we also have added a new reference (Bruner et al. Nature Med 2016;22:1043). In our analyses we identified hypermutated sequences and performed analyses with and without these sequences, with very similar overall results (see Figure 3—figure supplement 1 and elsewhere). You correctly state that with our sequencing method with we cannot identify sequences with deletions outside of the p17gag target region. This is a limitation of our study that we address in the Discussion: “Because we sequenced a relatively short region of the HIV-1 genome, we could not reliably distinguish between replication-competent and defective viruses. We did not find evidence of evolution in proviral DNA sequences, but we cannot rule out the possibility that a small subset of viruses was replicating and remained undetected among the many defective viruses.” However, apart from this caveat, the phylogenetic analyses actually are aided by defective proviruses because these sequences do not further evolve after integration. A sentence has been added to the Discussion to point this out: “However, our analyses of the composition and turnover of the virus reservoir were aided by the fact that most of the viruses were defective because these proviruses acted as inert in vivo labels of CD4 cell clones.”

*2) Paragraph 2 in Introduction (on QVOA) is difficult to understand. First, none of the QVOA information is used later in the text. The study quantifies neither the number of defective viruses nor the size of the reservoir. Second, while this is a matter of scholarship, the statement about PHA partial stimulation is problematic. There is no evidence that PHA in culture results in partial T-cell activation. The cited paper, Ho et al. 2013 only shows that there is partial HIV reactivation in culture, not the mechanism of how it arises. Partial reactivation of HIV may be due to intrinsic properties of the virus and should also be mentioned (Razooky et al. 2015, Rouzine et al. 2015).*

We agree that it is not clear exactly why the QVOA underquantifies.

Therefore, we have rephrased the text in the Introduction (second paragraph). We have also added the references that reviewer mentions together with the recent Bruner paper mentioned above.

*3) A very strong part of the study is the comparison between HIV DNA post-ART and the HIV RNA pre-ART. However, there was not discussion or analysis of this. In our opinion, Figure 4 needs to be substantially 'fleshed out'. It is a little difficult to understand particularly panels A versus B. The low half-life of T cells before ART is noted, but then as in the subsection “Time for deposition of reservoir DNA sequences”, there were patients that showed 14% and 42% of their DNA sample matching the initial plasma RNA. Neither of these are small numbers given the half-lives (time period without ART shown in Figure 1) and given the previous panel showing most sequences came from just before ART. Can you explain this so that these apparently contradictory issues are resolved?*

We agree that this section of the Results (Time for deposition of reservoir HIV-1 DNA sequences) lacked clarity and important details. Therefore we have extensively rewritten this section. We have also completely revised Figure 4 as well as added supplementary figures to this figure.

4) A more rigorous comparison to the data in Lorenzo-Redondo et al. (possibly in a new Discussion section) would be extremely helpful. The counter argument that will in all likelihood be used to rebut and limit this work as definitive is the classic "lack of data" argument (i.e., demonstrating that changes in the viral sequence occurred as a result of viral evolution is typically easier since there can be presence of signal in the sequence that appears to show such changes, as in Lorenzo-Redondo et al; in contrast, lack of data is always less conclusive because, for several reasons, this signal may have been missed, and the authors state this in the text. However, by comparison to the Lorenzo-Redondo study, the data here has more patients and time points but does not have multiple organs. It would be helpful if it is pointed out how much evolution was observed in Lorenzo-Redondo et al. PBMC data, in how many patients, and then show how unlikely or likely it would be for them to have not seen it.

Lorenzo-Redondo et al. estimated evolutionary rates for blood and lymph samples separately and found comparable rates in both compartments. Furthermore, they estimated substantial migration with mixing times smaller than 6 months between these compartments and state “We deduce that viral lineages in blood are derived from replicating virus in lymph nodes..”. Thus, we should be seeing evolved sequences in PBMC if substantial evolution had happened in lymph nodes and contributed a measurable fraction of HIV-1 DNA in PBMC. Our PBMC data derive from many more samples over a much longer time span than the data by Lorenzo-Redondo et al. Therefore, it is very unlikely that we could have missed a temporal signal of the magnitude Lorenzo-Redondo et al. report. We now include a supplementary figure (Figure 3—figure supplement 2) that indicates the expected patterns of root-to-tip divergence based on the rate estimates by Lorenzo-Redondo et al.

*If Lorenzo-Redondo et al. see it in PBMC data and the results here do not, why would that be? Also, in the second paragraph of their Discussion, the authors point out that Lorenzo-Redondo et al. report a higher than normal rate of evolution. In that paper however, different work was cited claiming their rates were normal. I think the authors should expand on this and specifically point out references and rates in the presence and absence of ART.*

Lorenzo-Redondo estimate rates between 0.6−1.0×10^-3^ substitutions per month and compare these rates to those reported in Lemey, P., Rambaut, A. & Pybus, O. G. AIDS Rev. 8, 125140 (2006). This paper estimated a rate between 0.5 and 0.8 × 10^-3^ per month. However, these estimates were for the C2-V5 region of gp120 using data by Shankarrappa et al., 1999. It is well-known that gag and pol evolve more slowly than gp120. Recent estimates suggest rates that are 3 to 10-fold lower (Figure 1 in Alizon and Fraser, Retrovirology 2013 or Zanini et al. 2016) than those estimated for C2V5.

As requested we have expanded the discussion of possible explanations for the opposing finding by Lorenzo-Redondo et al. and us (Discussion paragraphs #2-4). We have also added a supplementary figure (Figure 3—figure supplement 3) that illustrates how analyses of samples obtained too early after start of therapy may give a false impression of rapid evolution.

*5) Related to the above point, why does the analysis not account for clonal expansion? The authors should have done this analysis (see Maldarelli et al., 2014 or Cohn et al., 2015) to estimate the extent of expansion rather than leave that to speculation (even for a single patient where they assumed this was the top contribution). This would have taken out a major source of possible confusion. What if the sample prep or methodology are somehow dominated by clonal expansion which is why they don't capture any variation?*

With our data we cannot perform the type of analysis that Maldarelli et al. and Cohn et al. performed. They analyzed integration sites and thereby obtained molecular tags for each integrant/clone. We chose to analyze p17gag because it was part of our plasma HIV RNA dataset, in contrast to the extreme ends of the LTRs, and because it had sufficient genetic diversity to allow reliable comparison between plasma RNA and PBMC DNA HIV populations. We have expanded the discussion of the pros and cons of our sequence method and data (Discussion, paragraph #7).

As pointed out in the revised Discussion, several lines of evidence suggest that clonal expansion has not affected our results in a qualitative manner. Clonal expansion would result in undue over-representation of a some sequences. To control against such expansion, we repeated the analysis in Figure 3 and Figure 4 counting every haplotype only once, regardless of how many reads with that sequence were recovered (see supplements to these figures). By ignoring the number of times a sequence was sampled, effects of clonal expansion and PCR biases are removed. The results on the root-to-tip distances and the likely origin of the sequences are largely unchanged. Furthermore, data by Cohn et al. show only a limited decrease of single integrations following the start of ART (from about 70% to 50%, Figure 3).

*6) Figure 4: it is noted that most proviral sequences were assigned to a sample just before treatment. How many were assigned and how many could not be assigned by patient?*

We observed a total of 604 reads (out of 980,000 reads that began with the HIV specific primer) that could not be assigned to a patient sample. Such aberrant reads were observed in 10 out of 30 samples and the majority of these reads mapped to the human genome. This has been clarified in the Methods section (subsection “Sequencing and read processing”).